# NoiseAR: AutoRegressing Initial Noise Prior for Diffusion Models

## Abstract

Diffusion models have emerged as powerful generative frameworks, creating data samples by progressively denoising an initial random state. Traditionally, this initial state is sampled from a simple, fixed distribution like isotropic Gaussian, inherently lacking structure and a direct mechanism for external control. While recent efforts have explored ways to introduce controllability into the diffusion process, particularly at the initialization stage, they often rely on deterministic or heuristic approaches. These methods can be suboptimal, lack expressiveness, and are difficult to scale or integrate into more sophisticated optimization frameworks. In this paper, we introduce NoiseAR, a novel method for AutoRegressive Initial Noise Prior for Diffusion Models. Instead of a static, unstructured source, NoiseAR learns to generate a dynamic and controllable prior distribution for the initial noise. We formulate the generation of the initial noise prior's parameters as an autoregressive probabilistic modeling task over spatial patches. This approach enables NoiseAR to capture complex spatial dependencies and introduce learned structure into the initial state. Crucially, NoiseAR is designed to be conditional, allowing text prompts to directly influence the learned prior, thereby achieving fine-grained control over the diffusion initialization. Our experiments demonstrate that NoiseAR can generate initial noise priors that lead to improved sample quality and enhanced consistency with conditional inputs, offering a powerful, learned alternative to traditional random initialization. A key advantage of NoiseAR is its probabilistic formulation, which naturally supports seamless integration into probabilistic frameworks like Markov Decision Processes and Reinforcement Learning. This integration opens promising avenues for further optimizing and scaling controllable generation for downstream tasks. Furthermore, NoiseAR acts as a lightweight, plug-and-play module, requiring minimal additional computational overhead during inference, making it easy to integrate into existing diffusion pipelines.

## 1 Introduction

Recent breakthroughs in generative modeling, particularly with the advent of Diffusion Models (DMs) (Ho et al., 2020; Song et al., 2020a;b; Rombach et al., 2022; Podell et al., 2023), have revolutionized data synthesis, achieving unprecedented levels of fidelity and diversity, especially in image generation. These models achieve this by learning to reverse a gradual noise injection process, starting from a simple random noise sample – typically drawn from an isotropic Gaussian distribution (Ho et al., 2020) – and progressively refining it into a coherent data sample. While highly successful for unconditional generation, the practical utility of DMs in real-world scenarios heavily relies on the ability to control the generation process to produce outputs with specific desired attributes or according to explicit instructions. This capability is indispensable for tasks like text-driven content creation, complex image manipulation, and generating data with predefined structural or semantic characteristics.

Significant research efforts have been dedicated to making diffusion models controllable. Much of this work has focused on steering the generative process after the initial noise is sampled. Common strategies involve conditioning the denoising network throughout the reverse steps, using techniques like Classifier Guidance (Dhariwal & Nichol, 2021), Classifier-Free Guidance (Ho & Salimans, 2021; Nichol et al., 2021), or leveraging cross-attention mechanisms with conditional inputs (Rombach et al., 2022; Ramesh et al., 2021; Saharia et al., 2022a; Ramesh et al., 2022; Chefer et al., 2023;

Figure 1: Comparison of our autoregressive generation of initial noise (A) with existing approaches based on the refinement or mapping of initial noise (B). NoiseAR can generate initial noise from scratch without the need for any uncontrollable, more primitive noise.

Peebles & Xie, 2023b; Chen et al., 2023). Other methods manipulate the sampling path or apply objectives/constraints during the later stages of diffusion or related generative flows (Lipman et al., 2022; Liu et al., 2022; Karras et al., 2022; Albergo & Vanden-Eijnden, 2023; Albergo et al., 2023; Song & Dhariwal, 2024; Song et al., 2023; Geng et al., 2024; Lu & Song, 2025; Yang et al., 2024), including carefully designed noise schedulers (Nichol & Dhariwal, 2021; Chen, 2023; Lu et al., 2022) which govern the denoising dynamics. While these methods are effective at guiding how the denoising path unfolds, the generative process fundamentally begins with the initial noise. The potential of influencing the final output by injecting structured, controllable information right at this foundational starting point remains relatively underexplored compared to methods focusing on the later stages or process dynamics. Existing attempts to manipulate the initial state are limited, often relying on simple deterministic mappings (Fig. 1B) (Eyring et al., 2024; Ma et al., 2025; Zhou et al., 2024) or heuristic rules (Guo et al., 2024; Xu et al., 2025). Critically, these approaches still need to rely on uncontrollable gaussian for refining, which fail to model a flexible, probabilistic distribution over the initial noise conditioned on control, restricting their expressiveness and hindering integration with powerful probabilistic optimization frameworks.

In this paper, we explore this underexplored potential by proposing NoiseAR, a novel framework designed to learn a controllable, probabilistic prior distribution specifically for the initial noise of diffusion models. Unlike standard unstructured noise or deterministic initial state manipulations in Fig. 1B, NoiseAR (Fig. 1A) leverages the power of Autoregressive (AR) modeling (Van Den Oord et al., 2016; Van den Oord et al., 2016; Parmar et al., 2018; Chen et al., 2020; Li et al., 2024a) to capture complex spatial dependencies and define a conditional probability distribution over the initial noise grid. This allows NoiseAR to generate a structured, conditioned initial state distribution (e.g., mean and variance of Gaussian) from which samples can be drawn, offering a fundamentally new way to inject control and structure into the diffusion process right from its inception.

A key advantage of NoiseAR is its ability to model and provide access to the full probability distribution of the initial noise given the control signal, rather than merely outputting a single sample or a deterministic transformation. The probabilistic nature of our learned initial prior makes NoiseAR uniquely compatible with probabilistic optimization and decision-making paradigms like Markov Decision Processes (MDPs) and Reinforcement Learning (RL) (Sutton et al., 1998). This opens up new avenues for optimizing complex, high-level conditional generation objectives by learning to control the parameters of the initial noise distribution, leveraging the established power of frameworks integrating generative models with RL for planning and control (Hafner et al., 2019).

To our knowledge, NoiseAR is the first method to utilize autoregressive probabilistic modeling to learn a controllable initial noise prior for diffusion models, specifically designed to provide a learned, structured probabilistic starting point. We validate the effectiveness of NoiseAR in enabling enhanced controllable generation through comprehensive experiments with negligible computation. Our main contributions are summarized as follows:

- We propose NoiseAR, the first framework utilizing AR modeling to learn a controllable probabilistic prior distribution over the initial noise of diffusion models.

- We demonstrate that NoiseAR enables enhanced controllable generation by providing a learned, structured initial state distribution.

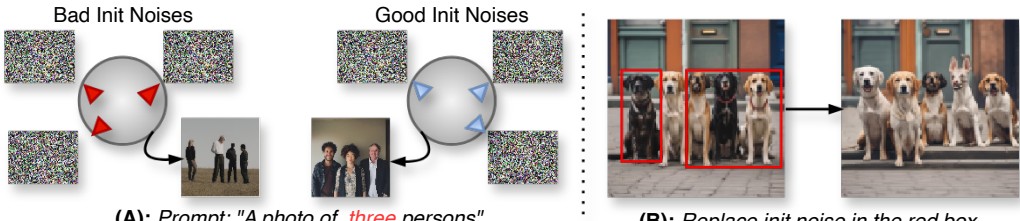

**(A):** *Prompt: "A photo of three persons"* **(B):** *Replace init noise in the red box*

Figure 2: Observation of Initial Noise in Diffusion Models. (A): Not all initial noise vectors yield desirable results; sampling in the vicinity of a "bad" noise vector consistently produces unfavorable outcomes, whereas sampling near a "good" one reliably generates high-quality results. (B): Localized edits to the noise vector (e.g., within the red bounding box) correspond to distinct, block-level modifications in the generated output.

- We highlight the unique advantage of NoiseAR's probabilistic nature, which facilitates seamless integration with probabilistic optimization frameworks like MDP/RL for future work on optimizing controllable diffusion generation.

## 2 METHOD

We brief our motivations. Fig. 2 presents our two novel observations regarding the initial noise in diffusion models, which motivate our approach:1) **Quality of Initial Noise Matters** (Fig. 2A): Not all initial noise vectors yield desirable results. This suggests that the latent space is locally consistent and learning a structured prior over the initial noise can significantly improve generation quality. 2) **Localized Control via Noise Patches** (Fig. 2B): This observation indicates a strong spatial correspondence between the noise map and the image canvas, motivating a patch-based autoregressive model to achieve fine-grained, controllable generation.

### 2.1 PROBLEM FORMULATION AND AUTOREGRESSIVE PRIOR

#### 2.1.1 PRELIMINARIES: DIFFUSION MODELS AND INITIAL NOISE

Diffusion Models (DMs) operate through a two-step process: a fixed forward diffusion process that gradually adds noise to data, transforming a data sample $\mathbf{z}_0$ into a pure noise sample $\mathbf{z}_T$ over $T$ steps; and a learned reverse denoising process that transforms the noise $\mathbf{z}_T$ back into a data sample $\mathbf{z}_0$. The reverse process, used for generation, starts from an initial noise $\mathbf{z}_T$, typically sampled from a simple, fixed prior distribution, most commonly the isotropic Gaussian distribution $p(\mathbf{z}_T) = \mathcal{N}(\mathbf{0}, \mathbf{I})$. While the standard practice of using unstructured Gaussian noise is simple and effective for unconditional generation, it provides no inherent mechanism to control the attributes, structure, or semantics of the final generated output from the very beginning.

#### 2.1.2 PROBLEM FORMULATION

Instead of relying on a fixed, unstructured Gaussian prior $p(\mathbf{z}_T)$, our goal is to learn a **controllable probabilistic prior distribution** over the initial noise tensor $\mathbf{z}_T \in \mathbb{R}^{C \times H \times W}$ (where $C, H, W$ are channels, height, and width) conditioned on a given control signal $\mathbf{c}$. Formally, we aim to learn the conditional probability distribution $P(\mathbf{z}_T | \mathbf{c})$. This learned distribution $P(\mathbf{z}_T | \mathbf{c})$ replaces the standard $p(\mathbf{z}_T)$, allowing us to sample a *structured* and *conditioned* initial noise $\mathbf{z}_T$ that is specifically tailored to the desired control $\mathbf{c}$, thereby influencing the diffusion process from its absolute start.

#### 2.1.3 AUTOREGRESSIVE PRIOR

To effectively model the complex dependencies and structure within $\mathbf{z}_T$ and its relationship with the control signal $\mathbf{c}$, we leverage the power of autoregressive (AR) modeling, applied at the patch level. This approach factorizes the joint probability distribution of $\mathbf{z}_T$ into a product of conditional probabilities over its constituent patches, ordered sequentially.

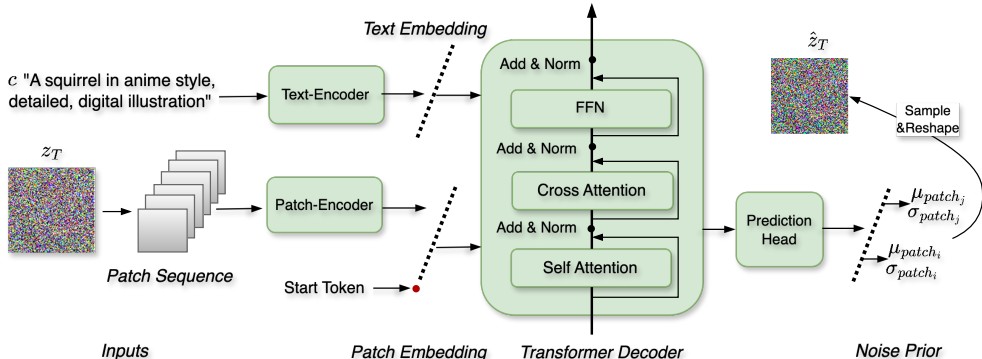

Figure 3: Overall Architecture of NoiseAR. During training, paired data $(\mathbf{z}_T, \mathbf{c})$ is used. The input noise $\mathbf{z}_T$ is processed by first dividing it into non-overlapping patches, flattening them, and projecting them into patch embeddings. The text-prompt $\mathbf{c}$ is processed by a separate text encoder to produce conditioning text embeddings. A learnable Start Token embedding (red dot) is prepended to the sequence of patch embeddings, and positional encodings are added to the entire sequence to form the input tokens for the Transformer Decoder. The Transformer Decoder stack processes this token sequence. Within each layer, masked multi-head self-attention captures dependencies among preceding tokens (patches and Start Token), enforcing the autoregressive property. Multi-head cross-attention integrates the conditioning vectors from the control signal. The final hidden states of the Transformer Decoder are fed to a Prediction Head, which outputs the parameters (e.g., mean $\mu_{patch}$ and log-variance $\log(\sigma^2_{patch})$) defining the conditional distribution for the next patch in the sequence. During inference, this process is used autoregressively to sample patch by patch based on the control signal $\mathbf{c}$, generating a conditioned $\hat{\mathbf{z}}_T$.

First, the 3D noise tensor $\mathbf{z}_T \in \mathbb{R}^{C \times H \times W}$ is spatially divided into $M = (H/P) \times (W/P)$ non-overlapping patches (Dosovitskiy et al., 2021; Peebles & Xie, 2023a; Ma et al., 2024), where $P$ is the patch size. These patches are then linearized into a 1D sequence of patches $\mathbf{Z}_T = [\mathbf{Z}_{T,1}, \mathbf{Z}_{T,2}, \dots, \mathbf{Z}_{T,M}]$ following a predefined ordering (a raster scan order by default). Each patch $\mathbf{Z}_{T,j}$ is itself a tensor containing $K = P \times P \times C$ elements.

Using this sequence of patches, the conditional probability distribution $P(\mathbf{z}_T|\mathbf{c})$ can be factorized autoregressively as:

$$P(\mathbf{z}_T|\mathbf{c}) = \prod_{j=1}^{M} P(\mathbf{Z}_{T,j}|\mathbf{Z}_{T,<j}, \mathbf{c})$$

where $\mathbf{Z}_{T,j}$ denotes the $j$-th patch in the sequence, and $\mathbf{Z}_{T,<j} = [\mathbf{Z}_{T,1}, \dots, \mathbf{Z}_{T,j-1}]$ represents all preceding patches in the defined order.

NoiseAR is designed to learn the parameters of these conditional distributions $P(\mathbf{Z}_{T,j}|\mathbf{Z}_{T,<j}, \mathbf{c})$ for each patch position $j$, conditioned on the previously processed patches and the control signal $\mathbf{c}$. Specifically, for each patch $\mathbf{Z}_{T,j}$, the model predicts parameters (mean and variance) for $K$ independent Gaussian distributions, conditioned on $\mathbf{Z}_{T,<j}$ and $\mathbf{c}$. The core AR dependency is maintained between sequential patches, allowing the model to build up spatial dependencies across the image. By modeling the distribution patch by patch in this sequential manner, the AR approach allows NoiseAR to capture dependencies between regions and learn a structured prior over $\mathbf{z}_T$.

## 2.2 NoiseAR Model Architecture

The NoiseAR model is designed to parameterize the conditional probability distributions $P(\mathbf{Z}_{T,j}|\mathbf{Z}_{T,<j}, \mathbf{c})$ derived from the patch-level autoregressive factorization of $P(\mathbf{z}_T|\mathbf{c})$. As illustrated in Figure 3. Our architecture is based on the powerful Transformer Decoder framework, well-suited for sequential data modeling with attention mechanisms. The model takes the control signal $\mathbf{c}$ and the sequence of previously processed noise patches (represented as tokens) as input and outputs the

parameters defining the probability distribution for the next patch in the sequence. The architecture consists of several key components:

### 2.2.1 Input Tokenization and Embedding

The raw input for the autoregressive model is constructed from the control signal $\mathbf{c}$ and the sequence of noise patches derived from $\mathbf{z}_T$.

**Noise Patching and Linearization:** As defined previously, the $C \times H \times W$ noise tensor $\mathbf{z}_T$ is first divided into non-overlapping patches of size $P \times P \times C$. These patches are then flattened into vectors and arranged in a predefined sequential order, forming a sequence of $M_{patches}$ patch vectors.

**Patch Embedding Layer:** Each flattened patch vector is projected into a higher-dimensional embedding space using a linear layer that maps the patch features ($\mathbb{R}^{P \times P \times C}$) to a token embedding vector ($\mathbb{R}^{D_{model}}$), where $D_{model}$ is the dimensionality of the model.

**Start Token Embedding:** A special, learnable vector ($\mathbb{R}^{D_{model}}$) is prepended to the sequence of patch embeddings. This Start Token serves as an initial input to the model, allowing it to generate the first patch's distribution based only on the control signal $\mathbf{c}$ and contextual information learned through this special token.

**Positional Encoding:** Transformers are inherently permutation-invariant, meaning they do not intrinsically understand the order of tokens in a sequence. To inject information about the spatial/sequential position of each patch (and the Start Token) within the overall grid structure, we add positional encodings to the token embeddings. These can be fixed or learned vectors (we use sinusoidal functions by default), added element-wise to the patch and Start Token embeddings before feeding them into the Transformer layers.

The resulting input sequence of tokens for the Transformer consists of the Start Token embedding followed by the patch embeddings of the noise sequence, totalling $M_{patches} + 1$ tokens.

### 2.2.2 Transformer Decoder Blocks

The core of NoiseAR is a stack of Transformer Decoder layers. Each layer typically comprises a masked multi-head self-attention block, a multi-head cross-attention block, and a position-wise feed-forward network. These layers process the sequence of input tokens (representing the Start Token and the noise patches) to build rich contextual representations.

**Masked Self-Attention:** This is the critical component enabling the autoregressive property at the patch level. For any given token position $j$ in the input sequence (corresponding to the Start Token or the $j$-th patch), the masked self-attention mechanism ensures that the token's representation can only attend to tokens at positions $k \leq j$. This prevents information leakage from future patches in the sequence, strictly adhering to the patch-level factorization $P(\mathbf{Z}_{T,j}|\mathbf{Z}_{T,<j}, \mathbf{c})$.

**Cross-Attention:** This block integrates the control signal $\mathbf{c}$ into the model. The control signal $\mathbf{c}$ is first processed (e.g., by a separate encoder network or simple projection layers) into a set of conditioning vectors. The Transformer sequence tokens (queries) attend to these conditioning vectors (keys and values), allowing the model to modulate its predictions based on the desired control. This ensures that the learned prior distribution is conditional on $\mathbf{c}$.

**Feed-Forward Network:** A standard two-layer feed-forward network with a non-linearity is applied independently to each token position after the attention blocks, enhancing the model's capacity.

### 2.2.3 Prediction Head

The final component is the prediction head, a stack of layers responsible for mapping the Transformer's output into the parameters of the conditional distribution for the next patch $\mathbf{Z}_{T,j}$. This head consists of a sequence of layers: a linear layer, followed by a GELU (Hendrycks & Gimpel, 2016) activation function, and a final linear layer. These layers take the hidden state from the Transformer's output corresponding to the position of the patch being predicted (a $D_{model}$-dimensional vector), and map it to an output vector of size $2 \times (P \times P \times C)$. These values represent the predicted means $\mu_{j,p_x,p_y,c}$ and log-variances $\log(\sigma^2_{j,p_x,p_y,c})$ for each of the $K = P \times P \times C$ individual elements $\mathbf{Z}_{T,j}[p_x, p_y, c]$ within the $j$-th patch. Consequently, the conditional distribution for the $j$-th patch, given the preceding

context and control signal, is a product of $P \times P \times C$ independent Gaussian distributions, one for each element:

$$P(\mathbf{Z}_{T,j}|\mathbf{Z}_{T,<j}, \mathbf{c}) = \prod_{p_x=1}^{P} \prod_{p_y=1}^{P} \prod_{c=1}^{C} \mathcal{N}(\mathbf{Z}_{T,j}[p_x, p_y, c]|\mu_{j,p_x,p_y,c}, \sigma_{j,p_x,p_y,c}^2)$$

As described in the problem formulation, each element $\mathbf{Z}_{T,j}[p_x, p_y, c]$ is sampled independently from its own conditional Gaussian distribution $\mathcal{N}(\mu_{j,p_x,p_y,c}, \sigma_{j,p_x,p_y,c}^2)$.

## 2.3 TRAINING OBJECTIVE

The model is trained to minimize the Negative Log-Likelihood (NLL) of the training data $(\mathbf{z}_T, \mathbf{c})$. Leveraging the autoregressive factorization over $M$ patches, the total NLL loss for a training pair is:

$$\mathcal{L}_{NLL}(\mathbf{z}_T, \mathbf{c}) = -\log P(\mathbf{z}_T|\mathbf{c}) = -\sum_{j=1}^{M} \log P(\mathbf{Z}_{T,j}|\mathbf{Z}_{T,<j}, \mathbf{c})$$

where $\mathbf{Z}_{T,j}$ is the $j$-th patch and $\mathbf{Z}_{T,<j}$ are preceding patches.

We use teacher forcing (Williams & Zipser, 1989) to produce training target. The model predicts the parameters $(\mu_{j,p_x,p_y,c}, \sigma_{j,p_x,p_y,c}^2)$ for all elements within patch $j$ based on ground truth $\mathbf{Z}_{T,<j}$ and $\mathbf{c}$. The loss for this step $j$ is computed as the sum of NLLs for all elements in the actual target patch $\mathbf{Z}_{T,j}$, where each element $z_{T,j}[p_x, p_y, c]$'s NLL is calculated using the specific predicted parameters $(\mu_{j,p_x,p_y,c}, \sigma_{j,p_x,p_y,c}^2)$ predicted for that element. Additionally, a 0.2-weighted reconstruction loss is calculated for the sampled data against the ground truth noise (GT), serving as an auxiliary loss.

## 2.4 INFERENCE AND SAMPLING

NoiseAR generates a novel $\hat{\mathbf{z}}_T$ autoregressively, patch by patch. Given $\mathbf{c}$ and previously sampled patches $\hat{\mathbf{Z}}_{T,<j}$, NoiseAR generates patch $\hat{\mathbf{Z}}_{T,j}$ for $j = 1, \ldots, M$ as follows:

1. Predict the parameters (means $\hat{\mu}_{j,p_x,p_y,c}$ and log-variances $\log(\hat{\sigma}_{j,p_x,p_y,c}^2)$) for each individual element $\hat{\mathbf{Z}}_{T,j}[p_x, p_y, c]$ within the target patch $\mathbf{Z}_{T,j}$, based on $\hat{\mathbf{Z}}_{T,<j}$ and $\mathbf{c}$. This results in $P \times P \times C$ pairs of $(\hat{\mu}, \log(\hat{\sigma}^2))$ values for the patch.

2. Sample each element $\hat{\mathbf{Z}}_{T,j}[p_x, p_y, c]$ independently from its corresponding predicted Gaussian distribution $\mathcal{N}(\hat{\mu}_{j,p_x,p_y,c}, \hat{\sigma}_{j,p_x,p_y,c}^2)$.

3. Append the sampled patch $\hat{\mathbf{Z}}_{T,j}$ to the sequence of generated patches.

Finally, the sequence of $M$ sampled patches is reshaped into the full noise tensor $\hat{\mathbf{z}}_T$.

# 3 EXPERIMENTS

## 3.1 EXPERIMENTAL SETUP

**Dataset:** To train our NoiseAR model, we constructed a dataset consisting of 100K pairs of (prompt, initial noise). We began by randomly sampling 100K prompts from the Pick-a-Pic training dataset (Kirstain et al., 2023), which contains a total of 1 million prompts. Using these prompts, we generated a synthetic initial noise using one-step Weak-to-Strong method (Bai et al., 2025), which applies one step forward and inversion to extract the corresponding initial noise vector $\mathbf{z}_T$ for each prompt. This process yielded our final training dataset of 100K $(\mathbf{c}, \mathbf{z}_T)$ pairs. Examples of training data can be found in Appendix E.2.

For evaluation, we utilized three test datasets: all 500 prompts from the Pick-a-Pic (Kirstain et al., 2023) test dataset, all 200 prompts from DrawBench (Saharia et al., 2022b), and all 553 prompts from GenEval (Ghosh et al., 2023). Further details regarding these datasets can be found in Appendix E.1.

**Downstream Diffusion Model(s):** We employed several pre-trained diffusion models as downstream generators, taking $\mathbf{z}_T$ sampled from NoiseAR prior. These included Stable Diffusion XL (Podell

Table 1: Performance Comparison of Initial Noise Generation Methods (NoiseAR, Standard Isotropic Gaussian Baseline, Golden Noise) across Downstream Diffusion Models and Benchmarks.

| | Downstream DM | Method | HPSv2↑ | AES↑ | Pick Score↑ | Image Reward↑ | CLIP Score(%)↑ | MPS(%)↑ |
|---|---|---|---|---|---|---|---|---|
| **DrawBench Results** | SDXL | Standard | 26.78 | 5.52 | 46.31 | 52.74 | 83.34 | 44.29 |
| | | Golden Noise | 27.47 | 5.52 | 53.53 | 57.49 | 83.30 | 52.83 |
| | | NoiseAR | 27.86 | 5.56 | 58.06 | 75.99 | 84.27 | 58.09 |
| | DreamShaper -xl-v2-turbo | Standard | 30.31 | 5.60 | 48.54 | 99.47 | 85.88 | 48.03 |
| | | Golden Noise | 30.18 | 5.59 | 51.45 | 97.57 | 85.79 | 51.96 |
| | | NoiseAR | 31.02 | 5.61 | 53.58 | 107.91 | 86.62 | 56.08 |
| | Hunyuan-DiT | Standard | 29.09 | 5.75 | 50.67 | 90.88 | 82.32 | 50.39 |
| | | Golden Noise | 29.02 | 5.74 | 49.32 | 89.66 | 82.42 | 49.60 |
| | | NoiseAR | 29.51 | 5.76 | 52.65 | 92.51 | 82.47 | 52.03 |
| **Pick-a-Pic Results** | SDXL | Standard | 28.58 | 5.92 | 47.40 | 74.07 | 83.25 | 46.21 |
| | | Golden Noise | 29.04 | 5.94 | 52.59 | 85.57 | 83.69 | 53.78 |
| | | NoiseAR | 29.40 | 5.95 | 54.56 | 90.72 | 84.13 | 56.27 |
| | DreamShaper -xl-v2-turbo | Standard | 32.70 | 6.00 | 48.77 | 118.82 | 85.34 | 44.67 |
| | | Golden Noise | 32.70 | 6.00 | 50.05 | 117.65 | 85.25 | 48.97 |
| | | NoiseAR | 33.03 | 6.01 | 50.15 | 121.06 | 86.03 | 50.83 |
| | Hunyuan-DiT | Standard | 29.78 | 6.12 | 50.52 | 95.94 | 81.29 | 49.64 |
| | | Golden Noise | 29.81 | 6.10 | 49.37 | 97.70 | 81.39 | 50.35 |
| | | NoiseAR | 30.24 | 6.13 | 50.60 | 106.80 | 81.59 | 54.46 |
| **GenEval Results** | SDXL | Standard | 27.80 | 5.45 | 46.30 | 40.92 | 81.15 | 45.04 |
| | | Golden Noise | 28.30 | 5.47 | 53.69 | 58.12 | 81.83 | 54.95 |
| | | NoiseAR | 28.61 | 5.48 | 58.09 | 68.33 | 82.27 | 54.98 |
| | DreamShaper -xl-v2-turbo | Standard | 31.02 | 5.45 | 47.75 | 98.06 | 83.78 | 45.34 |
| | | Golden Noise | 30.77 | 5.46 | 52.24 | 99.19 | 84.16 | 53.52 |
| | | NoiseAR | 31.75 | 5.47 | 52.51 | 109.63 | 84.17 | 55.08 |
| | Hunyuan-DiT | Standard | 30.26 | 5.64 | 50.43 | 107.51 | 82.76 | 49.02 |
| | | Golden Noise | 30.23 | 5.65 | 49.56 | 107.50 | 82.76 | 50.98 |
| | | NoiseAR | 31.12 | 5.67 | 53.73 | 116.59 | 83.15 | 55.12 |

et al., 2023), DreamShaper-xl-v2-turbo (fine-tuned from SDXL Turbo (Sauer et al., 2024)), and Hunyuan-DiT (Li et al., 2024b). And we all used 50 denoising steps at inference time.

**Evaluation Metrics:** To evaluate the performance of our NoiseAR model, we employ a set of metrics assessing generated image quality and text alignment. We utilize human preference metrics (HPS v2 (Wu et al., 2023), PickScore (Kirstain et al., 2023), ImageReward (IR) (Xu et al., 2023)) that capture perceived quality and adherence based on human judgments. We also report the Aesthetic Score (AES) (Schuhmann et al., 2022) for general aesthetic quality, CLIPScore (Hessel et al., 2021) for text-image alignment, and the Multi-dimensional Preference Score (MPS) (Zhang et al., 2024), offering a more comprehensive assessment across various dimensions of human preference. More details regarding these evalution metrics can be found in Appendix E.1.

## 3.2 QUANTITATIVE AND QUALITATIVE RESULTS

We present the quantitative evaluation of our NoiseAR model in this section, comparing its performance against baseline methods and demonstrating the benefits of reinforcement learning fine-tuning.

**Comparison with Baselines:** Table 1 shows the performance comparison of using initial noise sampled from our learned NoiseAR distribution against the standard isotropic Gaussian distribution (baseline) and the recently proposed Golden Noise (Zhou et al., 2024) method. We evaluate performance across different downstream diffusion models and test sets using the metrics described in Section 3.1. We can see guiding the diffusion model with the initial noise distribution learned by NoiseAR consistently and significantly outperforms using the isotropic Gaussian distribution. This superior performance indicates that NoiseAR effectively captures more informative structural

Table 2: Performance Comparison with and without DPO on **DrawBench** Dataset Using NoiseAR.

| Downstream DM | Method | HPSv2↑ | AES↑ | Pick Score↑ | Image Reward↑ | CLIP Score(%)↑ | MPS(%)↑ |
|---|---|---|---|---|---|---|---|
| SDXL | NoiseAR | 27.86 | 5.56 | 58.06 | 76.00 | 84.27 | 58.09 |
| | NoiseAR-DPO | 27.87 | 5.57 | 58.12 | 76.20 | 84.22 | 58.42 |
| DreamShaper -xl-v2-turbo | NoiseAR | 31.02 | 5.61 | 53.58 | 107.91 | 86.62 | 56.08 |
| | NoiseAR-DPO | 31.24 | 5.62 | 54.26 | 112.58 | 86.62 | 56.48 |
| Hunyuan-DiT | NoiseAR | 29.51 | 5.76 | 52.65 | 92.51 | 82.47 | 52.03 |
| | NoiseAR-DPO | 29.42 | 5.77 | 53.06 | 93.27 | 82.12 | 52.17 |

information in the initial noise space compared to a structureless Gaussian prior. Furthermore, our method also achieves better results than Golden Noise with analogical data collection method, which similarly aims to predict initial noise. We attribute this improved performance to our more sophisticated probabilistic modeling approach, specifically the autoregressive prediction of the distribution for each patch, which enables better generalization.

**Reinforcement Learning Fine-tuning with DPO:** Thanks to the probabilistic prior distribution learned by NoiseAR, the process of sampling initial noise can be naturally formulated as a Markov Decision Process. This allows us to leverage reinforcement learning techniques to further optimize the learned distribution for improved image generation quality and alignment with human preferences. We demonstrate the effectiveness of this approach by applying Direct Preference Optimization (Rafailov et al., 2023) as an initial validation. Our DPO data preparation was designed for simplicity and efficiency. After training the initial NoiseAR model on the cold-start dataset 3.1, we used 2,000 randomly sampled prompts from Pick-a-Pic training dataset for inference. For each of these prompts, we generated 20 image samples through separate rollouts (each involving sampling initial noise from NoiseAR and then denoising with the downstream model). We then used the previously described evaluation metrics (merged from IR, PickScore, and MPS) to score the resulting set of images for each prompt. For each prompt, we identified the image with the highest score and the image with the lowest score among the generated samples. A preference pair, consisting of the highest-scoring image (designated as the preferred sample) and the lowest-scoring image (designated as the rejected sample), was constructed only if the difference between the highest score and the lowest score for that prompt exceeded a threshold of 3.0. This filtering process based on score difference resulted in a final dataset of 348 preference pairs. For training, we use only simple NLL loss. Table 2 presents the results after fine-tuning the NoiseAR model with DPO on these preference pairs. It shows that applying DPO further enhances the performance in our chose metrics compared to the NoiseAR model before fine-tuning. A key advantage of using NoiseAR for generating DPO preference data is its inherent probabilistic sampling property, which naturally yields diverse samples for the same prompt, thereby facilitating the creation of informative preference pairs. This *contrasts* with methods that rely on sampling from a fixed, *uncontrolled* Gaussian distribution for the initial noise or a deterministic initial noise generation process, making it harder to generate varied rollouts for a given input.

**Visual Comparison with Baselines:** Figure 4 presents a visual comparison It shows that images generated using the initial noise sampled from our learned NoiseAR distribution are visually more coherent and plausible compared to those generated using the standard isotropic Gaussian baseline and Non-AR Golden Noise. More critically, the text-image alignment, representing how well the generated image matches the input prompt, is significantly improved with NoiseAR. Furthermore, after applying reinforcement learning fine-tuning with DPO, the consistency between the generated image and the text prompt is further enhanced.

## 3.3 Ablation Studies

To understand the contribution of different components of our NoiseAR model and its efficiency, we conduct several ablation studies.

**Impact of Patch Size:** We investigate the effect of the spatial patch size ($P \times P$) used for splitting the noise tensor on NoiseAR's performance (Table 3a). Results show that performance generally increases with patch size, peaking at $32 \times 32$. The smallest $4 \times 4$ patch size yields the lowest scores,

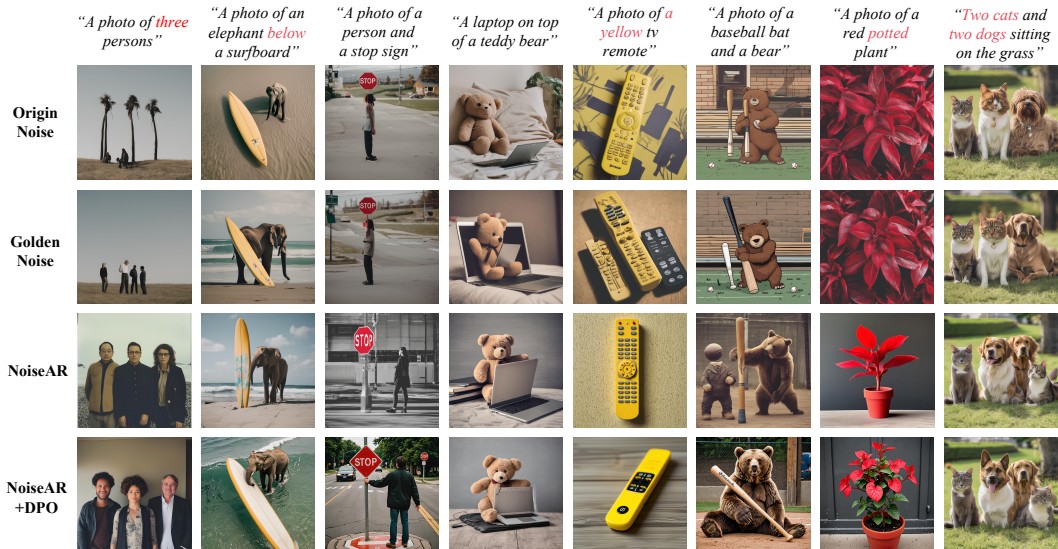

Figure 4: Visual Comparison of Image Generation Results using different Initial Noise Sources: Isotropic Gaussian (Baseline), Golden Noise, NoiseAR, and NoiseAR+DPO. The downstream DM and data are SDXL and DrawBench respectively. Note the improved visual coherence and text-image alignment with NoiseAR and NoiseAR+DPO.

Table 3: **Ablations.**

| Patch Size | CLIP Score | Image Reward |
|---|---|---|
| $4 \times 4$ | 83.59 | 65.99 |
| $8 \times 8$ | 83.88 | 68.68 |
| $16 \times 16$ | 84.13 | 75.04 |
| $32 \times 32$ | 84.27 | 76.00 |
| $64 \times 64$ | 84.17 | 74.36 |

(a) Effect of patch-size used for splitting noise.

| Decoder Layers | CLIP Score | Image Reward |
|---|---|---|
| 1 | 84.27 | 76.00 |
| 2 | 84.47 | 75.99 |
| 3 | 84.61 | 76.17 |
| 4 | 84.47 | 75.43 |
| 5 | 83.10 | 49.12 |

(b) Effect of layer stack number on Transformer Decoder.

| Head Layers | CLIP Score | Image Reward |
|---|---|---|
| 1 | 84.27 | 76.00 |
| 2 | 84.47 | 76.53 |
| 3 | 84.27 | 74.18 |
| 4 | 84.13 | 73.15 |
| 5 | 83.83 | 71.80 |

(c) Effect of layer stack number on Prediction Head.

likely due to the significantly increased autoregressive sequence length which raises training difficulty. Performance drops slightly for the $64 \times 64$ size. This suggests $32 \times 32$ provides the best trade-off between capturing contextual dependencies and managing sequence complexity.

**Impact of Network Depth:** We also ablate the depth of the core network components. Table 3b and Table 3c presents a comparison using different numbers of stacked layers for the Transformer decoder and the prediction head. The results indicate that noticeable performance improvements can be achieved even with a relatively small number of stacked layers. This demonstrates the robustness of our method, suggesting that significant gains in predicting a better initial noise distribution can be obtained without requiring an excessively deep architecture. In order to maintain high efficiency, this work uses only one layer by default, although the result is not the best.

**Efficiency Analysis:** We analyze the computational efficiency of our proposed method. As shown in Table 4, integrating NoiseAR introduces very little overhead to the overall inference process compared to the baseline diffusion model. The additional time cost is 0.2%, and the additional computational load is also negligible,

Table 4: Efficiency Analysis of NoiseAR.

| Model | GFLOPs | Speed (s/iter) |
|---|---|---|
| SDXL | 2600 | 15.00 |
| NoiseAR | 23.12 | 0.03 |

less than 1%. This high efficiency demonstrates that our method can be seamlessly integrated into existing diffusion pipelines as a plug-and-play module, highlighting its practicality and extendability.

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

Figure 5: Motivation for enhanced **end-to-end** controllable generation. We replace the uncontrolled isotropic Gaussian initial noise (top) with learned, prompt-conditioned autoregressive generation (bottom, highlighted by red arrows).

## A  MOTIVATION

**Motivation and Approach:** Traditional diffusion models, as illustrated in the top path of Figure 5, initiate the generation process by sampling initial noise from a standard isotropic Gaussian distribution. While simple, this approach introduces an element of randomness at the very start that is largely disconnected from the input text prompt, thus limiting the degree of end-to-end control achievable.

Our proposed NoiseAR model fundamentally addresses this limitation by replacing the uncontrolled isotropic Gaussian noise with a learned, structured initial noise distribution (bottom path in Figure 5). Critically, this initial noise is not random, but is generated autoregressively, directly conditioned on the input text prompt and a special start token. This key architectural difference, highlighted by the red arrows in Figure 5, represents a shift from an uncontrolled random initialization to a prompt-conditioned, learned generation process.

This novel approach offers two main advantages: firstly, providing a more informative and potentially text-aligned starting point can lead to improved visual quality and coherence in the final generated images. More significantly, by making the initial noise generation itself dependent on the text prompt, NoiseAR enables a much more enhanced and direct **end-to-end controllable generation** pipeline, where the prompt's influence extends to the very first step of the diffusion process, ensuring greater consistency between the input text and the generated image from the outset.

## B  RELATED WORK

### B.1  DIFFUSION MODELS AND CONTROLLABLE GENERATION

Diffusion Models (DMs) (Ho et al., 2020; Song et al., 2020a;b; Rombach et al., 2022; Podell et al., 2023) have become dominant in generative AI, excelling in synthesizing high-fidelity data, particularly images. DMs work by reversing a gradual noise-adding process, starting from a pure noise sample—typically drawn from a simple isotropic Gaussian distribution (Ho et al., 2020)—and iteratively denoising it into a coherent data sample. While powerful, the standard Gaussian initial noise provides no inherent structure or control handle for guiding the generated output from the outset. Significant research effort has been directed towards achieving controllable generation with diffusion models. Existing methods can broadly be categorized into several approaches:

1. Conditioning Mechanisms during Denoising: The most common approach is to integrate conditional information (e.g., text embeddings, class labels, spatial masks) directly into the diffusion model's denoising network throughout the reverse process. Techniques like Classifier Guidance(Dhariwal & Nichol, 2021), Classifier-Free Guidance (Ho & Salimans, 2021; Nichol et al., 2021), and cross-attention layers (Rombach et al., 2022; Ramesh et al., 2021; Saharia et al., 2022a; Ramesh et al., 2022; Chefer et al., 2023; Peebles & Xie, 2023b; Chen et al., 2023) allow steering

the generation towards outputs consistent with the given conditions by modifying the predicted noise or the estimated score function at each step. These methods effectively guide the 'path' of the diffusion process based on external input.

2. Initial State Manipulation: Some works have explored influencing the starting noise to affect the generation (Ma et al., 2025; Guo et al., 2024; Zhou et al., 2024). However, these approaches often rely on deterministic mappings from the condition to the initial noise (Ma et al., 2025; Zhou et al., 2024) or heuristic rules (Guo et al., 2024; Xu et al., 2025) for generating or modifying the initial state. Such deterministic or heuristic methods are limited in their expressiveness, may struggle to capture complex dependencies or structures inherent in a rich initial state, and critically, do not model a probability distribution over the initial noise, making them difficult to integrate seamlessly with probabilistic optimization frameworks.

3. Later Stage Manipulation: This category encompasses methods that alter the dynamics, speed, or specific steps of the diffusion process or analogous generative trajectory after the initial state is set(e.g., (Song et al., 2020a; Lipman et al., 2022; Liu et al., 2022; Karras et al., 2022; Albergo et al., 2023)). Noise schedulers, which define the sequence of noise levels $(\alpha_t, \sigma_t)$ over the diffusion steps within the diffusion framework, fall under this umbrella (Nichol & Dhariwal, 2021; Chen, 2023). They focus on how the denoising happens along the trajectory, rather than structuring what the starting point represents controllably.

Our work aligns with the "Initial State Manipulation" category but distinguishes itself by learning a structured, controllable probabilistic prior distribution for the initial noise, addressing the limitations of existing deterministic or heuristic approaches and opening avenues for integration with advanced probabilistic optimization.

## B.2    AUTOREGRESSIVE MODELING, LEARNED PRIORS, AND INTEGRATION WITH PROBABILISTIC FRAMEWORKS

Autoregressive (AR) models are powerful sequence models that learn complex joint distributions by factoring them into a product of conditional distributions. Their success is evident in high-quality text generation using Transformers (Sutskever et al., 2014; Vaswani et al., 2017; Radford et al., 2018; Devlin et al., 2019) and image generation tasks (Van Den Oord et al., 2016; Van den Oord et al., 2016; Parmar et al., 2018; Chen et al., 2020; Li et al., 2024a). AR models are particularly adept at capturing long-range dependencies and modeling structured data distributions sequentially, making them suitable for learning complex priors. The concept of learning rich prior distributions is well-established in other generative model families, such as Variational Autoencoders (VAEs) (Kingma et al., 2013). Replacing simple fixed priors (like isotropic Gaussians) with learned, flexible priors (e.g., using AR models or Normalizing Flows (Rezende & Mohamed, 2015)) has been shown to improve the generative capacity and sample quality of VAEs (Razavi et al., 2019). This underscores the potential benefits of learning a structured prior for a key component of a generative process. Furthermore, probabilistic modeling is a cornerstone of advanced decision-making and optimization frameworks, including Markov Decision Processes (MDPs) and Reinforcement Learning (RL) (Sutton et al., 1998). Algorithms in these fields often operate on or require access to probability distributions over states, actions, or outcomes. A generative model that provides a probabilistic representation, rather than just deterministic outputs, is thus naturally better positioned for integration into such frameworks, enabling tasks like policy optimization, value estimation, or model-based planning that rely on probabilistic transitions or outcomes (Hafner et al., 2019).

NoiseAR leverages the strengths of autoregressive probabilistic modeling by applying it to learn a controllable prior distribution for the initial noise of diffusion models. Unlike existing methods that deterministically generate initial states or use simple noise, NoiseAR learns a structured, conditional probability distribution over the initial noise. To our knowledge, NoiseAR is the first work to utilize autoregressive probabilistic modeling to learn a controllable initial noise prior specifically for diffusion models, offering a learned, structured starting point. Crucially, the probabilistic nature of our learned prior makes NoiseAR uniquely suited for seamless integration into probabilistic optimization frameworks like MDPs and RL, enabling future work on optimizing controllable diffusion generation through such methods.

## C  BROADER IMPACTS

Our work on learning a structured initial noise distribution for diffusion models significantly enhances the controllability and fidelity of text-to-image generation by improving text-image alignment. This offers considerable potential for positive applications, such as creating content that better reflects constructive human intentions and societal values, facilitating artistic expression, and aiding in educational or design processes. However, the increased ability to precisely control generated images also presents potential risks. The same technology that allows for better alignment with positive prompts can be used to generate harmful, misleading, or biased content more effectively when driven by malicious intent. This includes the potential for creating convincing misinformation, generating discriminatory imagery, or producing content that violates privacy or safety norms.

Therefore, responsible development, deployment, and careful consideration of ethical implications and potential misuse are paramount. Safeguards and policies to mitigate the generation and spread of harmful content will be increasingly important as models like NoiseAR enhance the capabilities of generative systems.

## D  TRAINING NOISEAR WITH REINFORCEMENT LEARNING

While the Negative Log-Likelihood (NLL) objective trains NoiseAR to accurately model the distribution of training data $\mathbf{z}_T$ at a patch level, it may not directly optimize for desired qualities of the final generated data sample $\mathbf{z}_0$. To address this, Reinforcement Learning (RL) offers a framework to optimize NoiseAR's initial noise generation for downstream criteria.

The NoiseAR model's autoregressive structure, which models the sequence patch by patch $\mathbf{Z}_{T,1}, \mathbf{Z}_{T,2}, \ldots, \mathbf{Z}_{T,M}$ based on previous patches and the control signal, lends itself to formulation as a Markov Decision Process (MDP). Each step $j$ in the autoregressive generation of a patch corresponds to a time step in the RL episode. The model's prediction of the conditional distribution $P(\mathbf{Z}_{T,j}|\mathbf{Z}_{T,<j}, \mathbf{c})$ defines the policy's output at each state. In this context, we frame the NoiseAR sampling process as an episodic Reinforcement Learning problem:

**Agent:** The NoiseAR model. Its "decision" at step $j$ is to define the conditional distribution for the next patch $\mathbf{Z}_{T,j}$ by predicting its parameters (means $\mu_{j,p_x,p_y,c}$ and log-variances $\log(\sigma^2_{j,p_x,p_y,c})$) for each individual element within the patch, for $p_x = 1, \ldots, P, p_y = 1, \ldots, P, c = 1, \ldots, C$.

**Environment:** Includes the partially generated sequence of patches, the control, the sampling process, the downstream Diffusion Model, and the reward function.

**State** ($s_j$)**:** At step $j$, the state is the input context for NoiseAR: the sequence of previously sampled patches $\hat{\mathbf{Z}}_{T,<j}$ and the control signal $\mathbf{c}$.

**Action** ($a_j$)**:** The action taken by the agent at step $j$ is sampling the entire patch $\hat{\mathbf{Z}}_{T,j}$. This patch is sampled by drawing each element $\hat{\mathbf{Z}}_{T,j}[p_x, p_y, c]$ independently from its corresponding predicted Gaussian distribution $\mathcal{N}(\hat{\mu}_{j,p_x,p_y,c}, \hat{\sigma}^2_{j,p_x,p_y,c})$.

**Policy** ($\pi$)**:** The NoiseAR model defines the policy $\pi(a_j|s_j)$, which is the conditional distribution $P(\hat{\mathbf{Z}}_{T,j}|\hat{\mathbf{Z}}_{T,<j}, \mathbf{c})$ for the next patch. This probability is the product of the probabilities of its individual elements, where each element's probability is determined by its element-specific predicted Gaussian:

$$P(\hat{\mathbf{Z}}_{T,j}|s_j) = \prod_{p_x=1}^{P} \prod_{p_y=1}^{P} \prod_{c=1}^{C} \mathcal{N}(\hat{z}_{T,j}[p_x, p_y, c]|\mu_{j,p_x,p_y,c}, \sigma^2_{j,p_x,p_y,c})$$

The log-probability $\log \pi(a_j|s_j)$ is straightforward to compute as the sum of the log-probabilities of all elements in the sampled patch, using the predicted parameters for each element.

**Episode:** Generating the complete sequence $\hat{\mathbf{z}}_T$ through $M$ sequential actions (sampling $M$ patches), followed by generating $\hat{\mathbf{z}}_0$.

**Reward** ($R$)**:** A scalar reward $R$ is assigned at the end of the episode, based on the quality of $\hat{\mathbf{z}}_0$. For e.g., we use score of ImageReward + (PickScore > 0.5 ) + (MPS > 0.5) to define the reward when collecting data for DPO.

The objective in this RL setup is to train the NoiseAR model (the policy $\pi$) to maximize the expected reward $E_\pi[R]$. Standard policy gradient methods can be adapted by using the computed log-probability of the sampled patch action $\log \pi(a_j|s_j)$, which is the sum of the log-probabilities of sampling each element independently from its predicted distribution.

# E EXPERIMENTS

## E.1 EXPERIMENTAL SETUP

### E.1.1 TRAINING DATASETS

**DrawBench** is a benchmark dataset specifically designed for the in-depth evaluation of text-to-image synthesis models. It was introduced by the Imagen to assess model performance comprehensively. DrawBench comprises a challenging set of prompts, often categorized to test various capabilities such as rendering colors accurately, counting objects, understanding spatial relationships, incorporating text into scenes, and generating images based on unusual interactions between objects. This structured suite of prompts allows for a rigorous comparison of different text-to-image models, helping researchers understand their strengths and weaknesses.

**Pick-a-Pic** is a large, open dataset focused on capturing real user preferences for images generated from text prompts. It was created by logging user interactions with a web application where users could generate images and then select their preferred output from a pair, or indicate a tie if neither was significantly better. The dataset contains over 500,000 examples covering 35,000 distinct prompts. A key advantage of Pick-a-Pic is that the preference data originates from genuine user choices rather than from paid crowd-sourcing, offering a more authentic reflection of user preferences. This dataset is instrumental in training preference prediction models like PickScore and is recommended for evaluating future text-to-image models.

**GenEval** is an object-focused framework and benchmark for evaluating the compositional alignment of text-to-image generative models. It aims to address limitations in holistic metrics like FID or CLIPScore by enabling a more fine-grained, instance-level analysis. GenEval evaluates properties such as object co-occurrence, position, count, and color by leveraging existing object detection models and can be linked with other discriminative vision models to verify specific attributes. The framework is designed to help identify failure modes in current models, particularly in complex capabilities like spatial relations and attribute binding, to inform the development of future text-to-image systems.

### E.1.2 TRAINING DETAILS

Training for NoiseAR model was conducted on a single NVIDIA A6000 GPU and completed within one hour. We trained the model for 10 epochs with a batch size of 40. The Adam (Kingma & Ba, 2015) optimizer was used, paired with a cosine learning rate scheduler for decay. The initial learning rate was set to 6.25e-5. The model architecture utilized a simplified structure where both the transformer decoder and the prediction head consisted of a single layer stack. A patch size of 32 was employed for speed and accuracy trade-off.

### E.1.3 EVALUATION METRICS

**Human Preference Score v2 (HPSv2)** is an advanced preference prediction model created by fine-tuning CLIP on the Human Preference Dataset v2 (HPD v2). This dataset is extensive, containing 798,090 human preference choices on 433,760 pairs of images, and is designed to mitigate potential biases found in earlier datasets. HPSv2 aims to align text-to-image synthesis with human preferences by predicting the likelihood of a synthesized image being preferred by users. It has demonstrated better generalization across various image distributions and responsiveness to algorithmic improvements in text-to-image models, making it a reliable tool for their evaluation.

**PickScore** is a CLIP-based scoring function trained on "Pick-a-Pic", a large, open dataset of real user preferences for images generated from text prompts. It has shown superhuman performance in predicting user preferences, achieving a high correlation with human judgments, even outperforming

expert humans in some tests. PickScore, especially when used with the Pick-a-Pic dataset's natural distribution prompts, enables a more relevant evaluation of text-to-image models than traditional standards like FID Heusel et al. (2017) over MS-COCO Lin et al. (2014). It is recommended for evaluating future text-to-image generation models due to its strong correlation with human rankings and its ability to assess both visual quality and text alignment.

**ImageReward** is a general-purpose human preference reward model specifically designed for evaluating text-to-image synthesis. It was trained on a substantial dataset of 137,000 expert comparisons, enabling it to effectively encode human preferences regarding aspects like text-image alignment and aesthetic quality. Studies have shown that ImageReward outperforms other scoring methods like CLIP and Aesthetic Score in understanding and aligning with human preferences. It serves as a promising automatic metric for comparing text-to-image models and selecting individual samples.

**Aesthetic Score (AES)** is a metric derived from a model trained on top of CLIP embeddings, typically with additional MLP (multilayer perceptron) layers, to specifically reflect the visual appeal or attractiveness of an image. It evaluates images based on factors like design balance, composition, color harmony, and clarity, providing a score (often 0 to 1) that quantifies how aesthetically pleasing an image is. This metric is used to assess the aesthetic quality of synthesized images, offering insights into how well they align with human aesthetic preferences.

**CLIPScore** is a reference-free metric that leverages the CLIP (Contrastive Language-Image Pre-training) model to evaluate the similarity or alignment between an image and a text description. It calculates the cosine similarity between the visual CLIP embedding of an image and the textual CLIP embedding of a caption in a shared embedding space. A higher CLIPScore, typically ranging from 0 to 100 (or -1 to 1 before scaling), indicates better semantic correlation between the image and the text. It has been found to correlate well with human judgment, particularly for literal image captioning tasks.

**Multi-dimensional Preference Score (MPS)** is the first preference scoring model designed to evaluate text-to-image models across multiple aspects of human preference, rather than a single overall score. It introduces a preference condition module built upon the CLIP model to learn these diverse preferences. MPS is trained on the Multi-dimensional Human Preference (MHP) Dataset, which contains 918,315 human preference choices across four dimensions: aesthetics, semantic alignment, detail quality, and overall assessment, covering 607,541 images generated by various text-to-image models. MPS calculates the preference scores between two images, where the sum of these two scores equals 1, and has shown to outperform existing methods in capturing these varied human judgments.

### E.1.4 DOWNSTREAM DIFFUSION MODELS

**Stable Diffusion XL (SDXL)** is a flagship open-source text-to-image generation model developed by Stability AI. It represents a significant advancement over previous Stable Diffusion versions, capable of producing higher-resolution images (typically 1024x1024 pixels) with enhanced photorealism, more intricate detail, and improved understanding of complex prompts. SDXL features a UNet backbone that is three times larger than its predecessors and often utilizes a two-stage pipeline: a base model generates initial latents, which can then be processed by a refiner model to add finer details and improve overall image quality. It also incorporates two text encoders (OpenCLIP-ViT/G and CLIP-ViT/L) to enhance prompt comprehension and supports features like image-to-image generation, inpainting, and outpainting. Due to its robust performance and open nature, SDXL is widely used in the image generation community and serves as a foundational model for many subsequent fine-tuned versions.

**DreamShaper-xl-v2-turbo** is a text-to-image generation model that has been fine-tuned from the Stable Diffusion XL (SDXL) base model, specifically stabilityai/stable-diffusion-xl-base-1.0. As suggested by "turbo" in its name, this model is optimized for faster image synthesis while aiming to maintain high-quality output, often with fewer sampling steps (e.g., 4-8 steps) and a low CFG scale (e.g., 2). The PDF document indicates that DreamShaper-xl-v2-turbo retains the high-quality image output characteristic of its predecessor and achieves quicker synthesis cycles due to its "turbo"

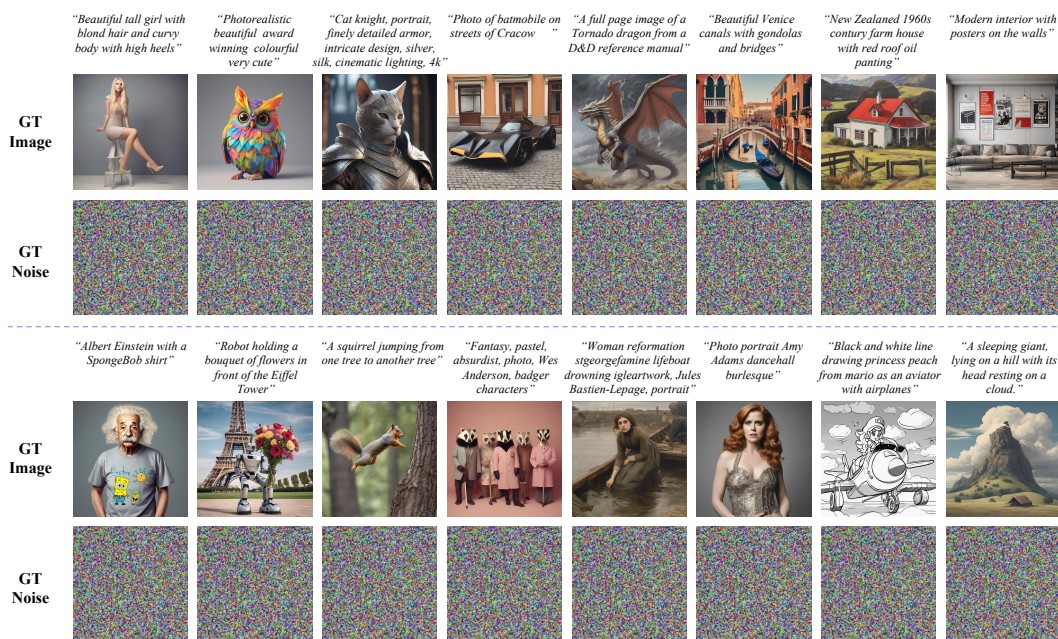

Figure 6: Examples of training data sampled from PickaPick for NoiseAR. Each column shows a text prompt, the corresponding image ($\mathbf{z}_0$), and the initial noise tensor ($\mathbf{z}_T$) generated by the diffusion model conditioned on the prompt. These representative examples are presented to illustrate the diverse inputs and targets used for training NoiseAR.

enhancement. It is described as excelling in various artistic styles, from photorealistic to anime and manga, with particular strengths in generating detailed human figures, sharp edges, and specific subjects like dragons. Models like DreamShaper are often tailored by creators like Lykon to excel in particular styles or to enhance efficiency for specific use cases.

**Hunyuan-DiT** is a text-to-image diffusion transformer model developed by Tencent Hunyuan. It is designed for fine-grained understanding of both English and Chinese text prompts. The model architecture features a diffusion transformer backbone operating in the latent space, leveraging a pre-trained Variational Autoencoder (VAE) for image compression. To encode text prompts, Hunyuan-DiT combines a bilingual (English and Chinese) CLIP with a multilingual T5 encoder. A notable feature mentioned in the PDF and search results is its ability to engage in multi-turn multimodal dialogues with users, allowing for iterative image generation and refinement based on conversational context. Tencent has also developed a comprehensive data pipeline and utilizes a Multimodal Large Language Model (MLLM) to refine image captions, enhancing the data quality for training and enabling the generation of images with high semantic accuracy, particularly for Chinese cultural elements.

## E.2 TRAINING DATA VISUALIZATION

To provide insight into the data used for training the NoiseAR model, we present a visualization of sixteen representative examples in Figure 6. As described in Section 3.1. NoiseAR is trained to model the conditional distribution $P(\mathbf{z}_T|\mathbf{c})$, where $\mathbf{z}_T$ is the initial noise tensor at the diffusion timestep $T$, and $\mathbf{c}$ is the conditioning signal (in our case, a text prompt). These examples showcase the variety of text prompts and the corresponding pairs of initial noise and final images used to teach NoiseAR how to generate appropriate initial noise priors conditioned on textual descriptions.

## E.3 DPO TRAINING DATA VISUALIZATION

Figure 7 presents 8 representative examples from the Pick-a-Pic dataset used for DPO training. As detailed in Section 3.2 , this dataset consists of preference pairs derived from outputs of the initial

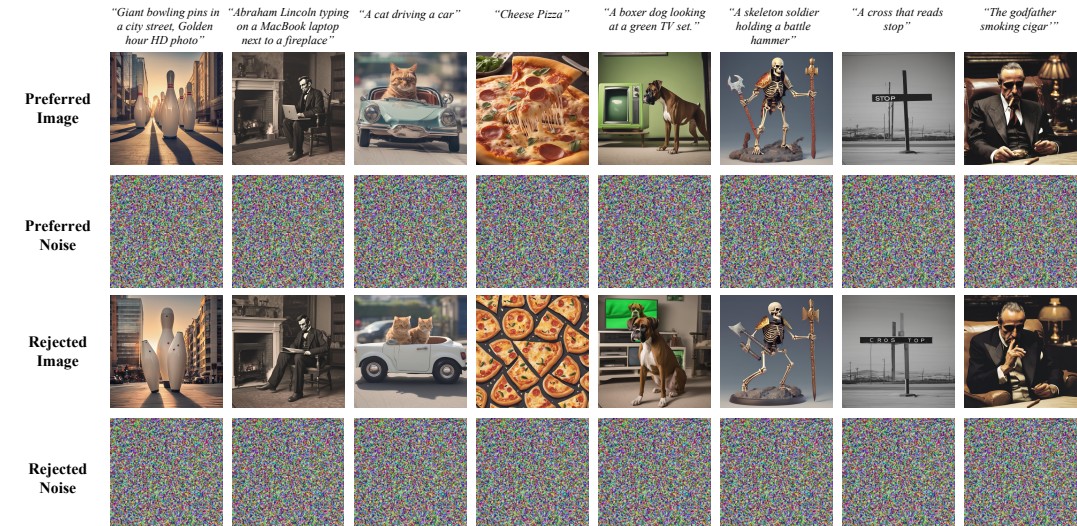

Figure 7: Examples of training data pairs for DPO. It displays the text prompt, the initial noise tensor ($\mathbf{z}_T^p$) that led to the preferred image, the preferred image ($\mathbf{z}_0^p$), the initial noise tensor ($\mathbf{z}_T^r$) that led to the rejected image, and the rejected image ($\mathbf{z}_0^r$). Eight representative pairs are shown to illustrate the structure and content of the DPO training dataset.

NoiseAR model, filtered based on score differences. For a given text prompt, it displays two generated outcomes: a preferred image and a rejected image, along with the specific initial noise tensors ($\mathbf{z}_T$) from which they were generated via the diffusion process. As indicated in the caption, each row thus comprises the text prompt, the initial noise and corresponding image for the preferred outcome, and the initial noise and corresponding image for the rejected outcome. Training with DPO on these pairs helps the NoiseAR model learn to assign higher probability to initial noise tensors like $\mathbf{z}_T^p$ that lead to preferred images ($\mathbf{z}_0^p$), and lower probability to tensors like $\mathbf{z}_T^r$ that result in rejected images ($\mathbf{z}_0^r$), conditioned on the same input prompt. These examples highlight the contrast between the initial noise inputs that produce subjectively (or metric-wise) better versus worse image results.

## F  LIMITATIONS

Despite the promising results achieved by NoiseAR in improving image generation quality and text-image alignment through a learned initial noise prior, our current work has several limitations that suggest avenues for future research. Firstly, our exploration of reinforcement learning fine-tuning was limited to using Direct Preference Optimization (DPO) as a proof-of-concept to demonstrate the potential benefits of optimizing the learned distribution. More sophisticated or alternative RL algorithms, such as Proximal Policy Optimization (PPO), could potentially yield further improvements. Furthermore, we did not investigate the scaling properties of NoiseAR or the effectiveness of learning the initial noise distribution with respect to model size, dataset size, or other relevant factors. Understanding these scaling laws would be crucial for assessing the method's performance and potential benefits at larger scales. Secondly, our method focuses on optimizing the *initial* noise distribution ($\mathbf{z}_T$) used to start the diffusion process. While theoretically orthogonal to techniques that modify the *intermediate* noise schedule or the denoising steps within the diffusion process, we did not conduct experiments to verify whether combining NoiseAR with such orthogonal techniques (e.g., advanced noise scheduling strategies or noise search methods applied at later timesteps) can lead to further synergistic improvements. Exploring these combinations could uncover additional performance gains. Finally, while our work focused exclusively on text-to-image generation, the core concept of learning a better prior distribution for the initial noise vector $\mathbf{z}_T$ is theoretically applicable to diffusion models across different modalities. This includes tasks like audio, video, and 3D generation, where diffusion models are increasingly used. Due to the scope of the current study, we were unable to explore the applicability and effectiveness of NoiseAR in these domains, which represents a significant area for future investigation.

## G  DECLARATION OF LLM USAGE

LLM is used only for writing, editing, or formatting purposes and does not impact the core methodology or originality of the research.

