# OpenReview forum: "NoiseAR: AutoRegressing Initial Noise Prior for Diffusion Models"
_ICLR.cc/2026/Conference — Submitted to ICLR 2026_

### Official Review · Reviewer_HRDB · 2025-10-28

**Soundness:** 3
**Presentation:** 2
**Contribution:** 2
**Rating:** 4
**Confidence:** 3

**Summary:**

NoiseAR reframes the role of initial noise in diffusion models: instead of sampling a static, isotropic Gaussian, it learns a prompt-conditioned, patch-wise autoregressive prior $P( \mathbf{z}_T | \mathbf{c})$ with a light-weight Transformer decoder that emits per-patch Gaussian parameters. This structured, probabilistic seed boosts text-image alignment and sample quality across SDXL, DreamShaper-xl-v2-turbo and Hunyuan-DiT while adding <1 % inference cost, and it can be further refined with RL-style Direct Preference Optimization. By making the very first latent variable optimizable and controllable, NoiseAR provides a plug-and-play pathway toward stronger, human-aligned generation for text-to-image and, potentially, video, audio and 3-D diffusion tasks.

**Strengths:**

The paper presents NoiseAR, a novel approach to learning a controllable, autoregressive prior for the initial noise in diffusion models.

---



- New Problem Framing: The paper reframes the role of initial noise in diffusion models—from an uncontrollable, fixed Gaussian sample to a learnable, structured latent variable. This is a conceptual shift that has not been fully explored in prior work.
- Creative Use of Autoregressive Modeling: While autoregressive models have been used in image and text generation, applying them to model the initial noise distribution of diffusion models is novel. The idea of predicting per-patch Gaussian parameters conditioned on text is both technically creative and practically powerful.
- Integration with RL Frameworks: The authors go beyond just modeling and show how their probabilistic formulation naturally fits into Markov Decision Processes (MDPs) and Reinforcement Learning (RL) setups, enabling future optimization via methods like Direct Preference Optimization (DPO).

- Strong Empirical Results: The method is evaluated across multiple diffusion models (SDXL, DreamShaper-xl-v2-turbo, Hunyuan-DiT) and multiple benchmarks (Pick-a-Pic, DrawBench, GenEval), showing consistent improvements in human preference metrics (PickScore, ImageReward, HPSv2, etc.).
- Ablation Studies: The paper includes thoughtful ablations on patch size, decoder depth, and prediction head layers, showing robustness and guiding practical implementation.
- Efficiency: The method adds <1% computational overhead, making it a plug-and-play module that is easy to integrate into existing pipelines.

---

NoiseAR is a creative, well-executed, and impactful contribution to the diffusion modeling literature. It challenges a long-standing assumption (that initial noise must be random) and provides a principled, efficient, and extensible alternative. The work is original in framing, high in quality, clear in exposition, and significant in both immediate and long-term implications.

**Weaknesses:**

Despite its creative approach and empirical robustness, NoiseAR still suffers from several practical shortcomings that constrain its immediate impact and broader future adoption.

The paper presents NoiseAR as the first learned, probabilistic prior over initial noise; however, deterministic or heuristic strategies for selecting noise have already been investigated:

- *Golden Noise for Diffusion Models: A Learning Framework*
- *Good Seed Makes a Good Crop: Discovering Secret Seeds in Text-to-Image Diffusion Models*
- *INITNO: Boosting Text-to-Image Diffusion Models via Initial Noise Optimization*

More importantly, the manuscript lacks a discussion of related work.

**Questions:**

The authors assume that a diffusion model fails to map Gaussian noise to a valid image because, for a *learned* diffusion model, the true initial noise distribution deviates from the Gaussian prior. Thus, they advocate *learning* the noise distribution instead. However, this raises a critical question: **how can the noise be learned when its ground-truth distribution is inherently unknown?**

1. **Training Data Construction**: Without access to the “correct” initial noise for real images, how is the training data for the noise model generated? If synthetic pairs (noise→image) are used, how are they guaranteed to reflect the *actual* noise distribution that the diffusion model implicitly inverts?

2. **Verification of Learned Noise**: Even if the proposed method produces high-quality images, this alone is insufficient evidence that the *noise distribution* has been accurately learned. After all, improvements could stem from unrelated inductive biases (e.g., smoother latent interpolation). What explicit validation confirms that the learned noise matches the “true” unknown distribution? For instance, are there diagnostics to rule out the possibility that the model merely learned a *different* heuristic (e.g., seed selection) unrelated to the underlying noise prior?

3. **Identifiability**: Given that the diffusion process is many-to-one (multiple noise inputs can map to similar images), how is the problem of learning the noise distribution *identifiable*? Without ground-truth noise samples, what prevents the model from learning arbitrary distributions that happen to yield plausible images?

---

> ### Author Response · Authors · 2025-11-24
> **Response to Reviewer HRDB**
>
> **W1: The paper presents NoiseAR as the first learned, probabilistic prior over initial noise; however, deterministic or heuristic strategies for selecting noise have already been investigated(e.g. Golden Noise, Good Seed, INITNO). Lack of discussion of these methods.**
>
> That is an excellent point.  Basically, there are fundamental differences between our approach and methods like Golden Noise or INITNO.
>
> - On a **methodological level**, these baselines operate by deterministically refining a pre-sampled noise from an N(0,1) distribution. Our NoiseAR method, however, autoregressively creates the noise from scratch, patch by patch. This AR process is a natural Markov Decision Process (MDP), a property that has been leveraged in language models and allows for a seamless integration with Reinforcement Learning (RL). In contrast, the deterministic, one-step refinement of Golden Noise and INITNO is less compatible with the exploratory nature of RL.
> - This methodological advantage is reflected in our **empirical results**. We have already included a comparison against Golden Noise in the original paper. Follow  your suggestion, we supplement the comparison and analysis of InitNO, Golden Noise and Good seed, in the **General Response section**, which confirms that our method consistently achieves superior performance.
>
> **Q1: Training Data Construction. how can the noise be learned when its ground-truth distribution is inherently unknown?**
>
> Our **main point** is that the objective is not to recover a ground-truth noise distribution, but to learn a better one. To this end, we employ a "weak-to-strong" data generation strategy proposed in [1], which is also used by Golden Noise[2]. Specifically, this is achieved using a "weak-to-strong[1]" data construction methodology: a noise from N(0,1) is first denoised with a high CFG scale, and then inverted back to the initial noise space using a lower CFG scale. The "weak-to-strong[1]" framework demonstrates that this inverted noise is superior to the original and embeds more information from the text prompt.
>
> Moreover, a key advantage of our method is the exploratory nature of the Autoregressive (AR) model. Even without knowledge of the ground-truth distribution, our AR model can proactively explore to find more effective noise samples. This capability to discover superior noise is precisely the rationale behind our Direct Preference Optimization (DPO) framework, as it guides the network's learning toward better outcomes.
>
> **Q2: Verification of Learned Noise.**
>
> As mentioned in Q1, our goal is to learn a better initial noise distribution, not to recover the ground-truth one. To disentangle the source of our improvements and demonstrate that they are not merely due to random perturbations, we conducted a new experiment against a random noise baseline, presented in the **General Response** section.
>
> Our method's performance is consistent superior to that of the random baseline. Crucially, we have used the same, **untuned seed** as the Golden Noise comparison in all experiments. This provides strong evidence that our model is not benefiting from a fortuitous random seed but is genuinely learning a better noise distribution.
>
> **Q3:Identifiability.**
>
> As mentioned in Q1, we can establish a valid learning objective by generating a superior noise target, a process we achieve using the same data construction methodology as Golden Noise[2] by using weak to strong data construction[1]. This is effective even in the absence of the ground-truth initial noise distribution. Moreover, Reinforcement Learning (RL) further ensures that our optimization is steered toward a more effective outcome.
>
> [1] Zigzag Diffusion Sampling: Diffusion Models Can Self-Improve via Self-Reflection, arXiv:2412.10891
>
> [2] Golden Noise for Diffusion Models: A Learning Framework, arXiv:2411.09502

---

### Official Review · Reviewer_3ktf · 2025-11-01

**Soundness:** 3
**Presentation:** 3
**Contribution:** 4
**Rating:** 6
**Confidence:** 3

**Summary:**

This submission proposed a new method to generate the initial noises that are conventionally sampled from a gaussian distribution for diffusion models.
Proposed NoiseAR is a Transformer-decoder-based autoregressive generator for 2D gaussian noises. NoiseAR is trained to maximize NNL on a collected noise-caption (here captions are of generated images from the corresponding noise) pair dataset, in other words NoiseAR is an autoregressive generative model of noises. Experiments using Pick-a-Pic, GenEval, and DrawBench text-to-image benchmarks, using NoiseAR for initialization is shown to be useful for multiple diffusion-based models.

**Strengths:**

- The idea of generating the initial noise is interesting and seems promising direction.
- The proposed method is plug-and-play, able to improved diffusion models.
- The experiments considers multiple aspects of the generated images including aesthetics (AES), human preference (HPSv2) and semantics (CLIP score). NoiseAR is shown to improve multiple metrics stably.

**Weaknesses:**

- Practical advantage over initial noise optimization (INO) is not well discussed. Is NoiseAR better than INO or different in applicable tasks?
- Other than generating the initial noise, a simpler dictionary-based noise collection method [a] was proposed. How does the proposed method compare with it?
  [a]The Lottery Ticket Hypothesis in Denoising: Towards Semantic-Driven Initialization
- Difference from the two-stage generation method: Having additional parameters in the NoiseAR module should be beneficial for generation by . From this viewpoint, NoiseAR may be a form of two-stage generation methods for example, the generator-refiner pattern seen in SDXL. The difference is that conventional two-stage generation is noise-to-image and image-to-image but NoiseAR is noise-to-noise and noise-to-image.
- The properties of the generated "noise" are not deeply discussed. Is it really noise, not one-step denoised latents?  If it is still a noise, how it is statistically different from the initial Gaussian?
- Construction of the training noise datasets {(Z_t, c)} is not clear: at first look, I thought that the noises that cause successful generation were collected using some filtering, but actually, no filtering or picking processes are involved?

**Questions:**

See weaknesses.

---

> ### Author Response · Authors · 2025-11-24
> **Response to Reviewer 3ktf**
>
> **W1: Practical advantage over initial noise optimization~(INO) is not well discussed**
>
> We thank the reviewer for the valuable suggestion. Assuming "INO" refers to InitNO, we have supplemented the **General Response** with a comparison and discussion showing that our method consistently outperforms it.
>
> This performance gain stems from a key architectural difference.
>
> 1. Our method leverages the text prompt to generate the initial noise from scratch patch by patch, ensuring that it is guided by the prompt from the very beginning. In contrast, methods like Golden Noise and INITNO perform a deterministic refinement on an initial noise that has **already been sampled from N(0,1)**, which limits their ability to utilize the prompt information at this crucial stage.
> 2. Our AR model's multi-step, probabilistic sampling is inherently more exploratory. Conversely, approaches like Golden Noise and INITNO are based on a deterministic refinement of a fixed, pre-sampled noise, making them less compatible with the exploratory requirements of Reinforcement Learning.
>
> **W2: Compared with: The Lottery Ticket Hypothesis in Denoising**
>
> We will add the comparison and discussion of the Lottery Ticket Hypothesis in the revised manuscript.
>
> We believe there are fundamental differences between our work and an approach based on the Lottery Ticket Hypothesis.
>
> **Methodologically:** The Lottery Ticket Hypothesis is a non-learning-based principle, whereas our method is entirely learning-based.
> **Empirically:** We simulated a comparable experiment by generating initial noise through **multiple random** samplings at the patch level. We found that the performance of patch-wise random sampling was similar to that of random sampling over the entire image. As shown in the table in the General Response, NoiseAR significantly outperforms this multi-seed random sampling baseline. Furthermore, in more **extensive local trials**, the best-performing instance of random patch sampling still did not match the performance of NoiseAR.
>
> **W3: Difference from the two-stage generation method: Having additional parameters in the NoiseAR module should be beneficial for generation by**
>
> This is an insightful question.
>
> - First, regarding computational cost, we found that adding an extra inference step to the diffusion model (with a computational cost equivalent to NoiseAR) yields negligible performance gains. This suggests that for SDXL, allocating computational resources to the initial noise generation phase is a more effective strategy.
> - Additionally, we would like to clarify that the characterization of NoiseAR as a "noise-to-noise" method is not entirely accurate. Our method generates the initial noise from scratch and does not require a pre-existing noise for refinement. To draw an analogy with language models, Autoregressive models are generative, creating new content sequentially. In contrast, bidirectional models, like the denoising process in diffusion, operate by refining a complete, pre-existing input. Therefore, our method generates the initial noise, rather than refining it. Conversely, methods such as Golden Noise and INITNO, which start with a sampled N(0,1) noise and then deterministically refine it, more closely fit the 'noise-to-noise' description.
> - Finally, we wish to emphasize that NoiseAR is orthogonal to image-to-image refinement methods. The two can be used in conjunction, and we anticipate that this combination could lead to further performance improvements.
>
> **W4: The properties of the generated "noise" are not deeply discussed**
>
> The output of NoiseAR remains a true Gaussian distribution, which distinguishes it from one-step denoised latents. Methodologically, NoiseAR is designed to predict only the parameters of a Gaussian distribution, whereas one-step denoised latents are more structured, image-like samples.
>
> This distinction is also statistically evident. We found that the mean of our generated Gaussians varies by up to ±0.3 across different patches. In contrast, standard N(0,1) noise has a uniform mean of 0 across all patches.
>
> **W5: Construction of the training noise datasets {(Z_t, c)} is not clear**
>
> We will add the description of the training data construction in the revised manuscript.
>
> Basically, we employ a "weak-to-strong" data generation strategy proposed in [3], which is also used by Golden Noise[4]. Specifically, this is achieved using a "weak-to-strong" data construction methodology: a noise from N(0,1) is first denoised with a high CFG scale, and then inverted back to the initial noise space using a lower CFG scale. The "weak-to-strong" framework demonstrates that this inverted noise is superior to the original and embeds more information from the text prompt.
>
> [1] The Lottery Ticket Hypothesis in Denoising,arXiv:2312.08872
>
> [2] InitNO,arXiv:2404.04650
>
> [3] Zigzag Diffusion Sampling: Diffusion Models Can Self-Improve via Self-Reflection,arXiv:2412.10891
>
> [4] Golden Noise for Diffusion Models,arXiv:2411.09502

---

### Official Review · Reviewer_3kvX · 2025-11-01

**Soundness:** 3
**Presentation:** 3
**Contribution:** 3
**Rating:** 6
**Confidence:** 5

**Summary:**

This paper addresses the lack of structure and external control in the traditional isotropic Gaussian initial noise of diffusion models by proposing NoiseAR, an autoregressive initial noise prior method. It learns a dynamic, controllable prior distribution for initial noise via patch-level autoregressive probabilistic modeling, using a Transformer decoder to split noise tensors into non-overlapping patches, predict Gaussian parameters with text prompts, and generate structured initial noise.

**Strengths:**

- First to apply autoregressive modeling to initial noise prior learning, filling the gap in structured control of diffusion starting state, with text prompts directly guiding noise generation.
- Validated across multiple datasets and models, with ablation studies clarifying parameter impacts and DPO proving iterability, ensuring reliable conclusions.

**Weaknesses:**

The author proposes a universal optimized solution for the initial noise distribution, and I have several questions regarding this:
- How does the size of the training set used by the author to train this model compare to that of the base model? If it is larger, wouldn’t the training cost be too high, and why not simply fine-tune the base model instead? If it is smaller, for samples that the base model has encountered but this model has not, could this initial noise optimization approach degrade the performance of the original base model?
- Regarding the lack of consistency between generated content and the prompt, can the inclusion of this module directly enhance the base model’s generative capability—meaning it enables the model to achieve what it previously could not—or does it merely increase the probability of generating content that aligns with the prompt?
- This method ultimately produces a distribution, and the initial noise is sampled from this distribution. Can every initial noise sampled from this optimized distribution generate results consistent with the prompt? If not, what is the success rate?
- Many base models initially use CLIP as the text encoder. However, CLIP is inherently insensitive to numerical concepts, such as "the number of people," which leads to a decline in generative capability. Using the CLIP score as a fine-grained alignment metric in this context seems biased.
- [1] has a very similar idea. Please give some comparison and discussion

[1] FIND: Fine-tuning Initial Noise Distribution with Policy Optimization for Diffusion Models. ACMMM 2024.

**Questions:**

- Can the inclusion of this module directly enhance the base model’s generative capability？
- Could this initial noise optimization approach degrade the performance of the original base model?
- Can every initial noise sampled from this optimized distribution generate results consistent with the prompt? If not, what is the success rate?
- Please provide a comparison with [1]

---

> ### Author Response · Authors · 2025-11-24
> **Response to Reviewer 3kvX**
>
> ****
>
> **W1 &  Q2:  How does the size of the training set compare to that of the base model? If smaller, could this initial noise optimization approach degrade the performance of the original base model**
>
> **Training data size:** As we introduced in the original manuscript, we began by randomly sampling 100K prompts from the Pick-a-Pic training dataset. This constitutes a very small dataset, especially when compared to the 5 billion text-image pairs in the LAION dataset commonly used for training Stable Diffusion models.
>
> Regarding the potential impact on the original model's performance, this is an excellent question. Interestingly, our model is initialized to predict a Gaussian distribution with a mean of 0 and a variance of 1. This means that for prompts or concepts not seen during our training, the model defaults to predicting the standard normal distribution, N(0, 1). Consequently, our method has a **negligible impact on the original model's performance**, as the base Stable Diffusion model also uses initial noise sampled from N(0, 1). However, for prompts similar to those in our training set, our model learns to predict a superior initial noise distribution, which leads to improved results.
>
> **W2 & Q1: Regarding the lack of consistency between generated content and the prompt, can the inclusion of this module directly enhance the base model’s generative capability—meaning it enables the model to achieve what it previously could not—or does it merely increase the probability of generating content that aligns with the prompt?**
>
> That is a very insightful question. Currently, NoiseAR is designed as a plug-and-play module to enhance Stable Diffusion without modifying the base model. The prospect of using our method to directly improve the intrinsic capabilities of the base model is a compelling research avenue, and it is one we intend to investigate in our future work.
>
> **W3 & Q3: This method ultimately produces a distribution, and the initial noise is sampled from this distribution. Can every initial noise sampled from this optimized distribution generate results consistent with the prompt? If not, what is the success rate?**
>
> Currently, established evaluation protocols do not yet include a standardized "success rate" metric or a corresponding benchmark. Consequently, we conducted our own human evaluation. Based on our sampled results, we found that approximately 68% of the generated images were perceived by human evaluators as being consistent with the prompt. While this result indicates that there is still considerable room for improvement, it is important to contextualize this performance.
>
> On existing, well-established metrics (such as human preference scores like ImageReward and PickScore, as well as Aesthetic Score), our method consistently outperforms both the standard isotropic Gaussian baseline and other approaches that refine it.
>
> **W4: Many base models initially use CLIP as the text encoder. However, CLIP is inherently insensitive to numerical concepts, such as "the number of people," which leads to a decline in generative capability. Using the CLIP score as a fine-grained alignment metric in this context seems biased.**
>
> That is an excellent observation. We also found in our experiments that CLIP is insensitive to numerical concepts. For this reason, we supplemented CLIP Score with other metrics to provide a more comprehensive evaluation of generation consistency. For example, metrics like PickScore and ImageReward can better assess consistency based on learned human preferences.
>
> **W5: Comparison with: FIND: Fine-tuning Initial Noise Distribution with Policy Optimization for Diffusion Models.**
>
> The primary distinctions between NoiseAR and FIND are as follows:
>
> 1. NoiseAR utilizes Autoregressive (AR) modeling to perform patch-wise, multi-step prediction. This formulates the task as a multi-step Markov Decision Process (MDP). In contrast, FIND operates as a single-step MDP.
> 2. A significant advantage of the patch-wise approach is that it allows each patch to be modeled by a different Gaussian distribution. This enables NoiseAR to have more flexible and granular control over the generation of different spatial regions. This is corroborated by the results in Table 3 of our original manuscript, where very large patch sizes were shown to be less effective than a 32x32 patch size.

---

### Official Review · Reviewer_ZfFt · 2025-11-05

**Soundness:** 2
**Presentation:** 1
**Contribution:** 1
**Rating:** 2
**Confidence:** 5

**Summary:**

This paper proposes to update the prior distribution of diffusion models as a mechanism for controlling generation. The central idea is to learn a structured prior for the initial noise, $ z_T $, enabling targeted sampling rather than controlling the entire denoising process. This is implemented using a patch-based, autoregressive model that is fine-tuned using a formulation based on RL. However, despite specific design choices, the paper's foundational idea is not new. The manuscript critically omits any mention of, or comparison with, existing methods that implement this exact "source-space guidance" concept.

**Strengths:**

Given the paper's overclaimed contributions and lack of comparison to the state-of-the-art, the strengths are limited:

1. The specific architectural choices, such as the patch-based autoregressive model, are technically sound.
2. The method demonstrates clear empirical improvements over the limited set of baselines chosen for comparison.

**Weaknesses:**

The submission's primary weakness is a critical failure of scholarship. It completely omits a mature and active line of research on the exact problem this paper claims to be exploring.The paper's entire narrative is built on the claim that influencing the "foundational starting point" ($ z_T $) "remains relatively underexplored". This is factually incorrect. A significant body of work is dedicated to this exact problem, invalidating the submission's core claims to novelty.The authors fail to cite or compare against numerous, highly relevant papers. The following is a non-exhaustive list of omitted prior and concurrent work:

1. Venkatraman, S., Hasan, M., Kim, M., Scimeca, L., Sendera, M., Bengio, Y., ... & Malkin, N. Outsourced Diffusion Sampling: Efficient Posterior Inference in Latent Spaces of Generative Models. In Forty-second International Conference on Machine Learning (ICML 2025).
2. Tang, Z., Peng, J., Tang, J., Hong, M., Wang, F., & Chang, T. H. Inference-Time Alignment of Diffusion Models with Direct Noise Optimization. In Forty-second International Conference on Machine Learning (ICML 2025).
3. Eyring, L., Karthik, S., Dosovitskiy, A., Ruiz, N., & Akata, Z. (2025). Noise Hypernetworks: Amortizing Test-Time Compute in Diffusion Models. arXiv preprint arXiv:2508.09968.
4. Kalaivanan, A., Zhao, Z., Sjölund, J., & Lindsten, F. (2025). ESS-Flow: Training-free guidance of flow-based models as inference in source space. arXiv preprint arXiv:2510.05849.
5. Wang, Z., Harting, A., Barreau, M., Zavlanos, M. M., & Johansson, K. H. (2025). Source-Guided Flow Matching. arXiv preprint arXiv:2508.14807.
6. Smith, H. D., Diamant, N. L., & Trippe, B. L. (2025). Calibrating Generative Models. arXiv preprint arXiv:2510.10020.
7. Om, K., Sim, K., Yun, T., Kang, H., & Park, J. (2025). Posterior Inference in Latent Space for Scalable Constrained Black-box Optimization. arXiv preprint arXiv:2507.00480. (used for Bayesian optimization)


This omission leads to several critical flaws:
1. The paper's claim to be the first RL-based approach, or to be "uniquely compatible" with RL , is false. For example, Venkatraman et al. ("Outsourced Diffusion Models") explicitly uses an RL objective (Trajectory Balance, related to continuous GFlowNets) to train its noise-space sampler, which also functions as a policy in an MDP.
2. The paper fails to benchmark against direct architectural competitors. Eyring et al.  ("Noise Hypernetworks") also proposes a structure-based approach to learn an amortized, conditional prior in the latent space. A comparison between the proposed autoregressive model and this hypernetwork approach is essential to validate the authors' design choices, but it is absent.
3. As a consequence of these omissions, the experimental validation is insufficient. By comparing only against weak baselines (e.g., the standard Gaussian prior ), the paper fails to demonstrate how its method performs against the actual state-of-the-art in this field. The main claims of the paper are, therefore, unsubstantiated.

**Questions:**

In addition to the Weaknesses, I would like to ask the authors to address the following question:

1. Please clarify the novelty of your work in light of the existing papers, especially "Outsourced diffusion sampling" and "Noise Hypernetworks" papers, both of which also propose learning a reward-driven, amortized prior in the noise space. Specifically, how is your RL-based formulation novel when  already uses an RL-based MDP (GFlowNet)?

---

> ### Author Response · Authors · 2025-11-24
> **Response to Reviewer ZfFt**
>
> **W1:The authors fail to cite or compare against numerous, highly relevant papers.**
>
> We are grateful for the reviewer's suggestions. We note that the majority of the cited literature represents **concurrent work**, with many papers appearing on arXiv only after our initial submission (e.g., [1][2] in Oct. 2025; [3][4]  in Aug. 2025; [5] in Jul. 2025).  Due to time constraints, we are unable to directly compare results of these methods. At the methodological level, the key novelty that distinguishes our method from these is our pioneering use of an Autoregressive (AR) approach for initial noise generation. Critically, this AR formulation is naturally a Markov Decision Process (MDP), allowing for a seamless and unique integration with Reinforcement Learning (RL).
>
> As recommended, we have updated the related works section to include a discussion of "Inference-Time Alignment of Diffusion Models with Direct Noise Optimization.[6]" and "Outsourced Diffusion Sampling[7]".
> We wish to reiterate that the key novelty distinguishing our work is the **Autoregressive (AR) modeling** of initial noise, where we sequentially generate the noise **patch by patch** from scratch. However, [6,7] are mainly rely on **refine** the initial noise sampled from N(0,1), which is not the same as our method.
>
> **W2: The paper's claim to be the first RL-based approach, or to be "uniquely compatible" with RL, is false.**
>
> We will clarify our use of "uniquely compatible." We are not the first to apply RL; rather, our novelty lies in the Autoregressive (AR) generation of initial noise. This AR framework is a natural Markov Decision Process (MDP), enabling a seamless integration with RL that is unique to this noise generation context.
>
> **W3: The paper fails to benchmark against direct architectural competitors(Noise Hypernetworks), As a consequence of these omissions, the experimental validation is insufficient.**
>
> Regarding "Noise Hypernetworks," it is a concurrent work ([3] arXiv: August 13, 2025), so a direct empirical comparison was not feasible. Instead, we benchmarked our method against similar approaches like InitNO and Golden Noise(please refer to General Response section, NoiseAR Consistent outperform these methods).
>
> The primary methodological difference is that these existing methods refine an initial noise sampled from an isotropic Gaussian (N(0,1)), whereas our NoiseAR directly generates a more expressive noise from its learned parameters patch by patch. Architecturally, NoiseAR leverages a causal transformer for patch-wise autoregressive generation. This design not only improves generative quality but, as a natural Markov Decision Process (MDP), it also enables a seamless integration with Reinforcement Learning (RL) frameworks.
>
> **Q1: Please clarify the novelty of your work in light of the existing papers.**
>
> As other reviewers have noted, our core contribution is the pioneering use of an Autoregressive (AR) model for initial noise generation. A key advantage of this formulation is that the AR process is a natural Markov Decision Process (MDP), allowing for seamless integration with Reinforcement Learning (RL), a principle well-established in language models. We see the further optimization of RL's exploratory power in this context as an avenue for future research.
>
> In contrast to "Noise Hypernetworks," which refines a pre-sampled isotropic noise, our method generates noise from the ground up in a patch-wise manner. Furthermore, when compared to the more generalized framework of "GlowNet[8]," our approach is specifically tailored to and experimentally validated for the text-to-image generation task. We will add this discussion to the related works section.
>
> [1] Calibrating Generative Models. arXiv preprint arXiv:2510.10020.
>
> [2] ESS-Flow: Training-free guidance of flow-based models as inference in source space. arXiv preprint arXiv:2510.05849.
>
> [3] Noise Hypernetworks: Amortizing Test-Time Compute in Diffusion Models. arXiv preprint arXiv:2508.09968.
>
> [4] Source-Guided Flow Matching. arXiv preprint arXiv:2508.14807.
>
> [5] Posterior Inference in Latent Space for Scalable Constrained Black-box Optimization. arXiv preprint arXiv:2507.00480. (used for Bayesian optimization).
>
> [6] Inference-Time Alignment of Diffusion Models with Direct Noise Optimization. In Forty-second International Conference on Machine Learning (ICML 2025).
>
> [7] Outsourced Diffusion Sampling: Efficient Posterior Inference in Latent Spaces of Generative Models. In Forty-second International Conference on Machine Learning (ICML 2025).
>
> [8] GFlowNet Foundations. arXiv:2111.09266

---

### Author Response · Authors · 2025-11-24
**General Response**

We would like to firmly express our gratitude to all reviewers for their insightful and constructive Comments, and appreciate the comments like  "The idea of generating the initial noise is interesting"(3ktf), "First to apply autoregressive modeling"(3kvX), "NoiseAR is a creative, well-executed, and impactful contribution to the diffusion modeling literature" (HRDB) and "Ablation studies ensuring reliable conclusions"(3kvX).

In accordance with the reviewer's suggestions, we have supplemented our experiments with a comparison to "InitNO[1]" and random noise using various seeds[2]. The results, presented in the subsequent two tables, indicate that our proposed method achieves superior performance over these baselines.

1. We theorize that the inferior performance of random noise is due to its inability to utilize information from the text prompt.
2. The "InitNO" approach refines an initial noise sampled from a standard normal distribution (N(0,1)), making its success contingent on the quality of this initial sample. However, if the starting noise is suboptimal, subsequent refinement struggles to correct it—a limitation analogous to diffusion models that also cannot easily recover from a poor initial noise, as highlighted in our motivation.  Conversely, our method employs an autoregressive (AR) process to create noise from scratch guided by the text prompt. This allows for a more effective integration of the text prompt, thereby generating a higher-quality initial noise from the start.

> Comparison with InitNO in GenEval benchmark. (Note: To ensure a fair and faithful comparison, we performed our evaluation on Stable Diffusion (SD). This is because the official implementation of InitNO was only evaluated on SD, with no results released for Stable Diffusion XL (SDXL)).
>

| Method | Backbone | HPSv2 | AES | Pick Score | Image Reward | CLIP Score | MPS |
| --- | --- | --- | --- | --- | --- | --- | --- |
| InitNO | SD | 23.49 | 4.73 | 23.81 | 33.71 | 76.51 | 22.06 |
| NoiseAR | SD | **24.03** | **5.10** | **24.02** | **35.68** | **77.22** | **23.64** |

> Comparison of random noise. All evaluation are conducted on the GenEval benchmark. Please note that the Golden Noise use a fixed seed of 5555, we just follow the same setting for fair comparison without any optimization for random noise.
>

| Method | Backbone | Seed | HPSv2 | AES | Pick Score | Image Reward | CLIP Score | MPS |
| --- | --- | --- | --- | --- | --- | --- | --- | --- |
| **NoiseAR** | SDXL | 5555 | **28.61** | **5.48** | **58.09** | **68.33** | **82.27** | **54.98** |
| Golden Noise | SDXL | 5555 | 28.30 | 5.47 | 53.69 | 58.12 | 81.83 | 54.95 |
| N(0,1) | SDXL | 5555 | 27.80 | 5.45 | 46.30 | 40.92 | 81.15 | 45.04 |
| N(0,1) | SDXL | 42 | 28.03 | 5.36 | 53.94 | 50.27 | 81.78 | 52.64 |
| N(0,1) | SDXL | 55 | 28.50 | 5.42 | 54.07 | 56.58 | 81.39 | 52.89 |
| N(0,1) | SDXL | 742 | 28.57 | 5.45 | 54.95 | 55.22 | 81.34 | 54.18 |
| N(0,1) | SDXL | 6239 | 28.00 | 5.40 | 50.66 | 45.28 | 81.05 | 50.95 |
| N(0,1) | SDXL | 9811 | 27.73 | 5.40 | 51.17 | 45.68 | 81.29 | 49.59 |

[1] InitNO: Boosting Text-to-Image Diffusion Models via Initial Noise Optimization, arXiv:2404.04650

[2] Good Seed Makes a Good Crop: Discovering Secret Seeds in Text-to-Image Diffusion Models, arXiv:2405.14828

---

### Author Response · Authors · 2025-12-03
**Summary of Response to Main Concerns**

We sincerely appreciate the insightful and constructive comments from all reviewers, as well as the time and effort dedicated by the Area Chairs and Senior Area Chairs.

In this rebuttal, we primarily address the key concerns raised by the reviewers:

**1. Novelty & Contribution**

- **General Recognition:** The majority of reviewers acknowledged the novelty of our work. Reviewers commended the method as the “First to apply autoregressive” (**3kvX**), “interesting, promising” (**3ktf**), and “New Problem Framing, creative use of AR” **(HRDB)**.
- **Differentiation from Prior Work:** Reviewer **(ZfFt)** expressed concern that the “**start point**” has been fully studied (**“mature and active”**) and noted a lack of citations for “**numerous, highly relevant papers.**”
    - **Our Response:** We clarify that our approach is fundamentally different from existing methods. We are the first to utilize Autoregressive (AR) models to generate the starting point. This offers unique advantages, including: **1) more flexible local generation** (patch-wise); **2) the ability to synthesize initial noise** (unlike methods such as GoldenNoise or InitNO, which require deterministic refinement from an existing noise from N(0,1) distribution); and **3) seamless integration with reinforcement learning.**
- **Regarding Citations:** regarding the missing citations mentioned by Reviewer **(ZfFt)**, we note that most of the recommended papers are concurrent works or were published after our submission. Therefore, we have prioritized adding comparisons with relevant works published prior to NoiseAR in the revision.

**2. Empirical Results**

- **Overall Performance:** On the whole, most reviewers affirmed our experimental results, citing “clear empirical improvements” (**ZfFt**), “ensuring reliable conclusions” (**3kvX**), the ability to “improve multiple metrics stably” (**3ktf**), and “Strong Empirical Results” (**HRDB**).
- **Robustness Analysis:** Reviewer (**HRDB**) raised a concern regarding whether the accuracy gains stem from different random initializations (e.g., random seeds). In our **General Response**, we demonstrate that NoiseAR performs significantly better than extensive random initializations. This analysis also directly addresses Reviewer **(3ktf)**’s inquiry regarding the "Lottery Ticket" hypothesis.

**3. Clarity & Thoroughness**

- **Ground-Truth Distribution:** Reviewer (**HRDB**) expressed concern that the ground-truth distribution of the initial noise is unknown. In our rebuttal, we have detailed that our data construction process aligns with that of GoldenNoise. We clarify that our objective is to acquire a superior initial noise to enhance generation, rather than to explicitly capture the ground-truth distribution.

Thank you again for your valuable efforts, which have greatly advanced our research.

---

### Meta-Review · Area_Chair_Xbgg · 2025-12-24

**Summary:**

This paper develops an approach for learning the right noise for a diffusion model given a context. The idea is that certain noise values lead to better samples and this could be learned. The noise value model is an auto regressive model.  Reviewers at onset were mixed in their initial assessment. The paper has potential, but issues in the related work and clarity around the actual training process, the sampling distribution of z and c in the negative log-likelihood make this paper need more before publication.

**Reviewer Concerns:**

Addressed:
- Questions about empirical results

Partially addressed:
- Related work (several). This were several pieces of related work that were raised by the reviewers. The authors have some discussion of these in the replies. There were also some comparisons added. I didn't feel like there was a sharp statement

Outstanding:
- Training data construction reviewers HRDB, 3ktf. This is a pretty serious concern. From my look at the paper, I agree with this. The authors responses do not clarify this enough to the point where it's clear how this process is done

**Reviewer Scores:**

ZfFt - novelty and related work were only partially addressed and they were confident. stay the same

3kvX - positive and remain

3ktf - had questions about related work. these may be amplified after reading ZfFt, so decrease

HRDB - negative and remain because the training data construction was not sharply answered

---

### Decision · Program_Chairs · 2026-01-26

Reject